# 3D Body-Wave Velocity Structure of the Southern Aegean, Greece

Andreas Karakonstantis [1,*] and Filippos Vallianatos [1,2]

1   Institute of Physics of the Earth's Interior and Geohazards, UNESCO Chair on Solid Earth Physics and Geohazards Risk Reduction, Hellenic Mediterranean University Research Center (HMURC), 73133 Chania, Crete, Greece; fvallian@geol.uoa.gr
2   Section of Geophysics-Geothermy, Department of Geology and Geoenvironment, National and Kapodistrian University of Athens, 15784 Zografou, Athens, Greece
*   Correspondence: akarakon@geol.uoa.gr; Tel.: +302107274285

**Abstract:** This study delves into the southern Aegean regionwhere the subduction of the oceanic Mediterranean lithosphere under the Aegean continental one takes place. This region is considered one of the most active ones in the eastern Mediterranean Sea due to intense tectonic movements in the Late Quaternary. More than 1200 manually revised events from 2018 to 2023 have been used in order to obtain the 3D structure of body-wave velocity and $V_P/V_S$ ratioto 80 km depth through earthquaketomography. A series of resolution tests have been performed and demonstrated fair resolution of the derived velocity structures in the area of interest. The derived anomalies of body-waves ($dV_P$, $dV_S$) and $V_P/V_S$ ratio provided important information about the southern Aegean regional tectonics and secondarily active faults of smaller scale (>20 km). The region is marked by significant low-velocity anomalies in the crust and uppermost mantle, beneath the active arc volcanoes. The seismicity related to the Hellenic Subduction Zone (HSZ) is connected to a low-angle positive anomaly of $V_P$ and $V_S$, correlated withthe observed intermediate-depth seismicity ($H \geq 40$ km) in this part of the study area. This result could be related to the diving HSZ slab.

**Keywords:** seismic tomography; lithospheric slab; southern Aegean; Hellenic Trench

## 1. Introduction

The southern Aegean, situated in Greece, boasts the southernmost part of the Hellenic arc, which commences in the area W-SW of Peloponnese and runs southward, dividing into the Pliny, Ptolemy, and Strabo trenches. The arc culminates to the east of Rhodes Island, branching into Anatolia and the south of Cyprus [1,2]. This region is prone to frequent seismic activity, owing to the convergence of the Eurasian and African tectonic plates, as substantiated by significant historical seismic events [3]. This movement is caused by the subduction of the oceanic Mediterranean lithosphere, which belongs to the Nubia plate under the Aegean continental one, with a speed of roughly 35 mm/year [4–10].

This study delves into the southern Aegean region (Figure 1), which has a significant portion of the overall recorded intermediate-depth seismic activity. This particular area is considered one of the most active ones in the eastern Mediterranean Sea due to intense tectonic movements during the Late Quaternary period, resulting in significant uplifts in local areas and the creation of extensional faults that run in roughly two directions: WNW-ESE and NNE-SSW [4,11–13]. In the Cretan Sea, just north of Crete, there are three primary tectonic depressions between the Southern Cyclades and Crete, Karpathos, and Rhodes. The main tectonic structures in the region are oriented in a NW-SE direction towards the west and a NE-SW direction towards the east, showcasing the curvature of the Hellenic arc. The broader area of Crete has a complex geodynamic structure where transpressional tectonics dominate the southern part of Crete, in contrast to normal and strike-slip forms on the shallow part of the crust, resulting in a heterogeneous stress field [14,15]. In the

northern area, between Santorini and Crete, there is an observation of extension in an E-W direction, while in the western part of Crete, N-forward shortening was seen [16] (Figure 2).

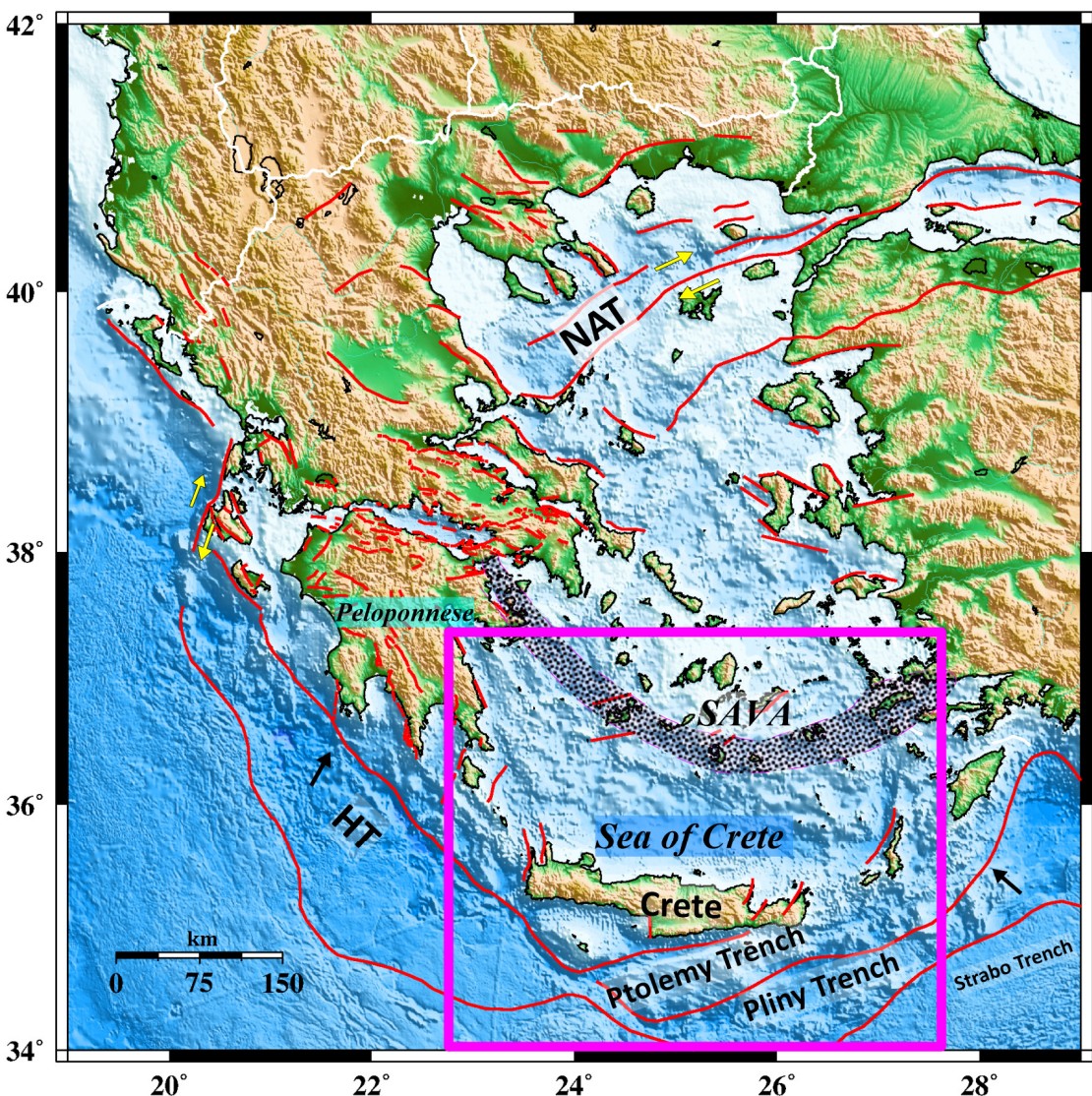

**Figure 1.** Main tectonic features in Greece and western Turkey. The rectangle in magenta color contains the study area. Abbreviations-HT: Hellenic Trench; NAT: North Aegean Trough; SAVA: South Aegean Volcanic Arc. Fault traces (red lines) derived by [9,17].

It is interesting to note that the seismic activity in Crete and the neighboring areas is mainly due to the geodynamics of the Aegean and Mediterranean plate convergence, with events of shallow depth (<25 km) characterizing it. However, the lack of uniform and optimal coverage with seismic stations on the island and the southern Aegean Sea and the absence of local velocity models are the main disadvantages in highlighting the earthquake distribution in areas near or along the Hellenic Trench (HT). Fortunately, recent seismic sequences have led to an improvement in the local station coverage, enriching the available catalogues. With improved 1D regional models and well-localized aftershocks recorded by dense seismic monitoring networks, the region can now provide a basis and baseline for more constrained tomographic studies. The recorded data improved the knowledge of the seismotectonic structures on the island and helped to provide a better 3D body-wave velocity model through seismic tomography, examining the correlation between seismicity and the activation of the local fault pattern.

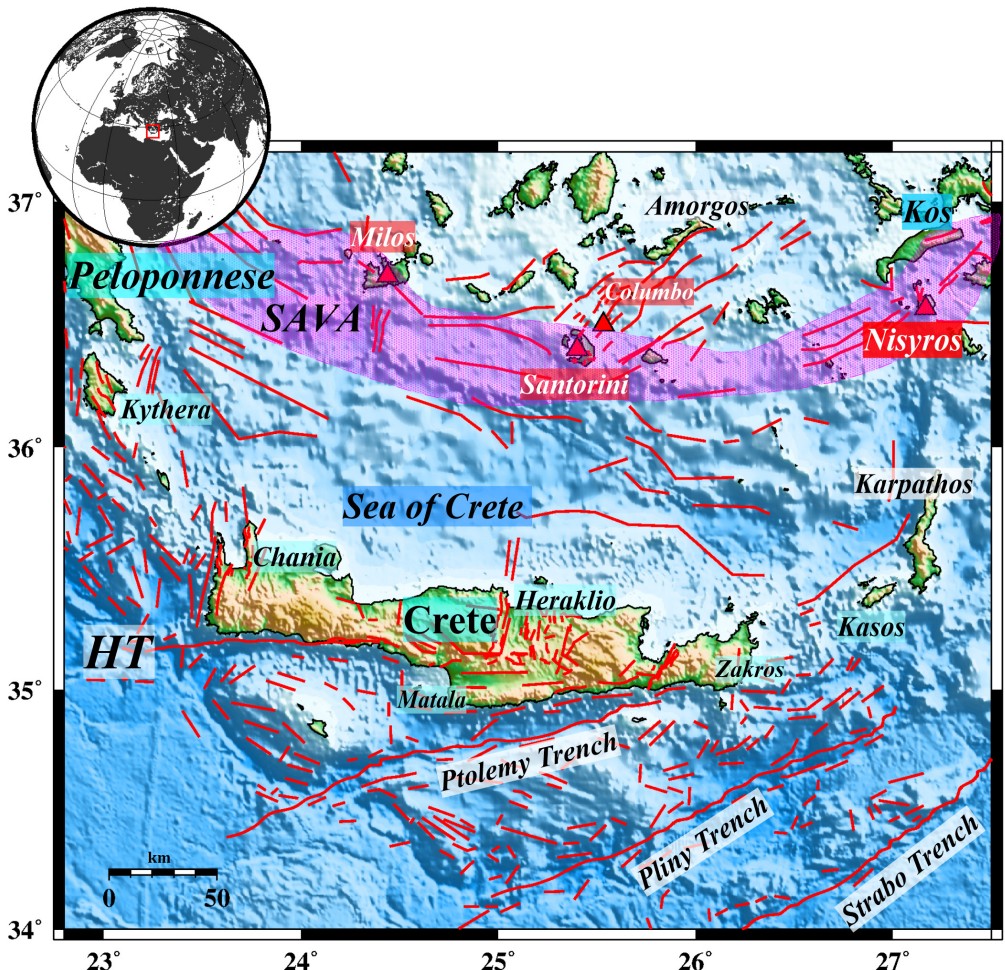

**Figure 2.** Main geological elements in the area of study. The purple shaded area contains the volcanic arc. The volcanic centers are noted in red triangles. Abbreviations-HT: Hellenic Trench; SAVA: South Aegean Volcanic Arc. Fault traces (red lines) derived by [17].

Seismicity studies since 1990 have revealed active Quaternary faults, major compressional structures (i.e., thrusts), and other geologic features in the back-arc basin north of Crete Island. The planes for the available focal mechanism solutions of offshore events near the Hellenic Trench south of Crete (Figure 3) show a direction of compression to be nearly NNE-SSW, while the onshore ones appear to have an almost ESE-WNW to N-S extension, which is associated with fractures along the main mapped faults [3,12,17–19]. Historical records indicate that three large shallow earthquakes occurred in this region, with the first one occurring southwest of Crete in 365 AD, the second one occurring near Rhodes in 1303, and the third one taking place on 16 February 1810, in the area between Crete and Rhodes [19,20]. The existence of permanent stations of the regional Hellenic Unified Seismological Network [21], close to the most significant events, with the nearest stations being Knossos (KNSS), Pefkos (PFKS) (Institute of Physics of the Earth's Interior and Geohazards of the Hellenic Mediterranean University Research Center; IFEGG-HMURC), and temporary ones [22,23] installed after large earthquakes, makes a significant contribution to the depth accuracy of the first hypocentral solution during the routine analysis in near real-time (http://www.geophysics.geol.uoa.gr/stations/gmaps3/leaf_stations.php?map=2&lng=en (accessed on 31 May 2023)).

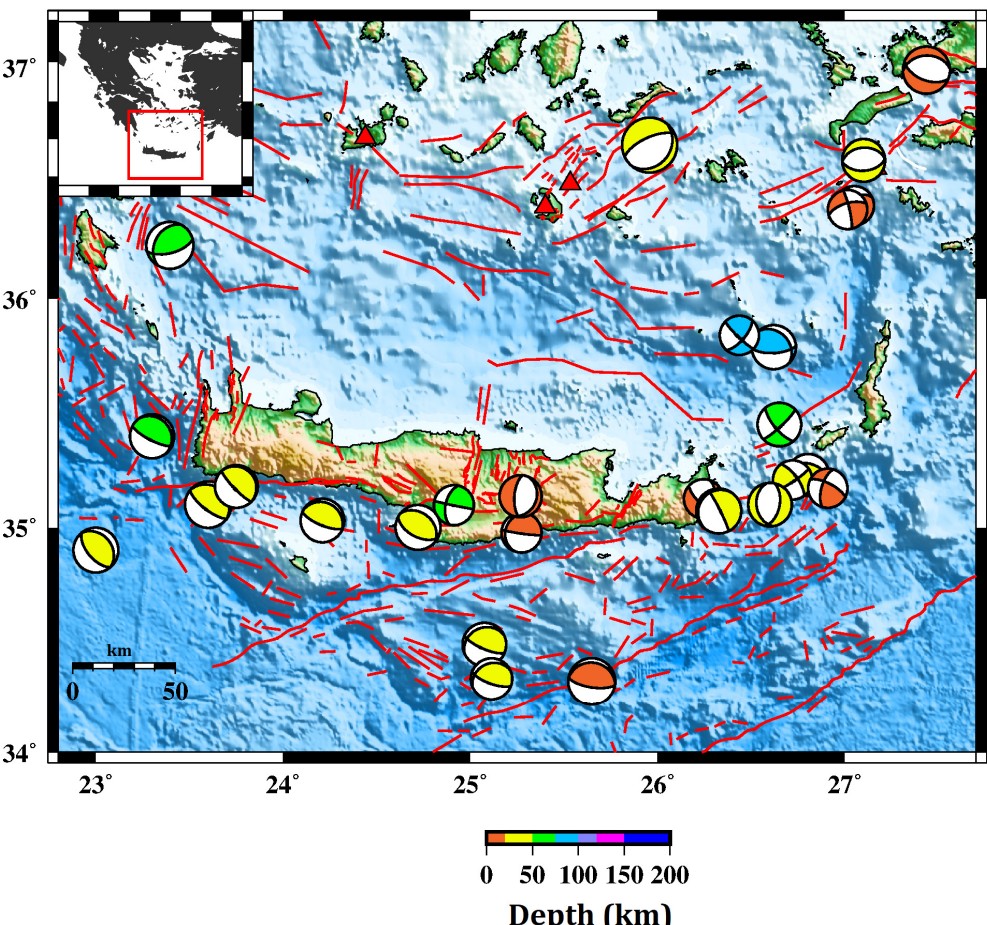

**Figure 3.** Focal mechanism solution of earthquakes with ML ≥ 5.5 during the instrumental time period [12,18,19]. Fault traces (red lines) derived by [17].

In the area of the Aegean Sea (Greece), there have been many studies conducted on a regional scale over the past forty years. Seismic velocity models have been obtained for various areas including western Greeceand the Aegean Sea [24–27]. However, it is worth mentioning that the lack of permanent seismic stations on the islands of the southern Aegean has made imaging of the crustal structure more challenging. In this work, we provide new 3D body-wave velocity models of the crust and uppermost mantle (down to 80 km) utilizing a new big collection of earthquake travel time data. These findings may aid in determining the relationships between surface geology and subsurface structures driven by complex geological processes.

## 2. Data and Method

The present study focuses on Crete and its neighboring regions in the southern Aegean Sea. Several permanent stations of the Hellenic Unified Seismological Network [22,23,28–30] operate in this area, complemented by 4 temporary stations (CRE1–4) of the Geodynamics Institute of the National Observatory of Athens (GI-NOA), between 27 September 27 and 15 June 2022, mainly contributing to the depth accuracy of the Arkalochori aftershock sequence for this period of time (Figure 4) [21,31–33]. The initial dataset comprises 1813 events from June 2018 to February 2023, with $M_L \geq 3.2$, recorded by at least 6 stations. Since 2007, continuous waveform recordings have been obtained from HUSN in realtime, comprising both 1-component (1 Hz) (THR2-9; Santorini Island Seismic Network operated by the Aristotle University of Thessaloniki) and 3-component (either broad-band or short-period) stations, while manual arrival-time picking is applied to obtain hypocentral locations using the SeisComP3 graphical user interface [34] and a custom regional 1D velocity model for the region

of the southern Aegean Sea [35]. In the first stage of the sequence analysis, hypocenters were located in near realtime, employing the Hypo71 single-event algorithm [36]. Then they were relocated running the HypoInverse code [37].

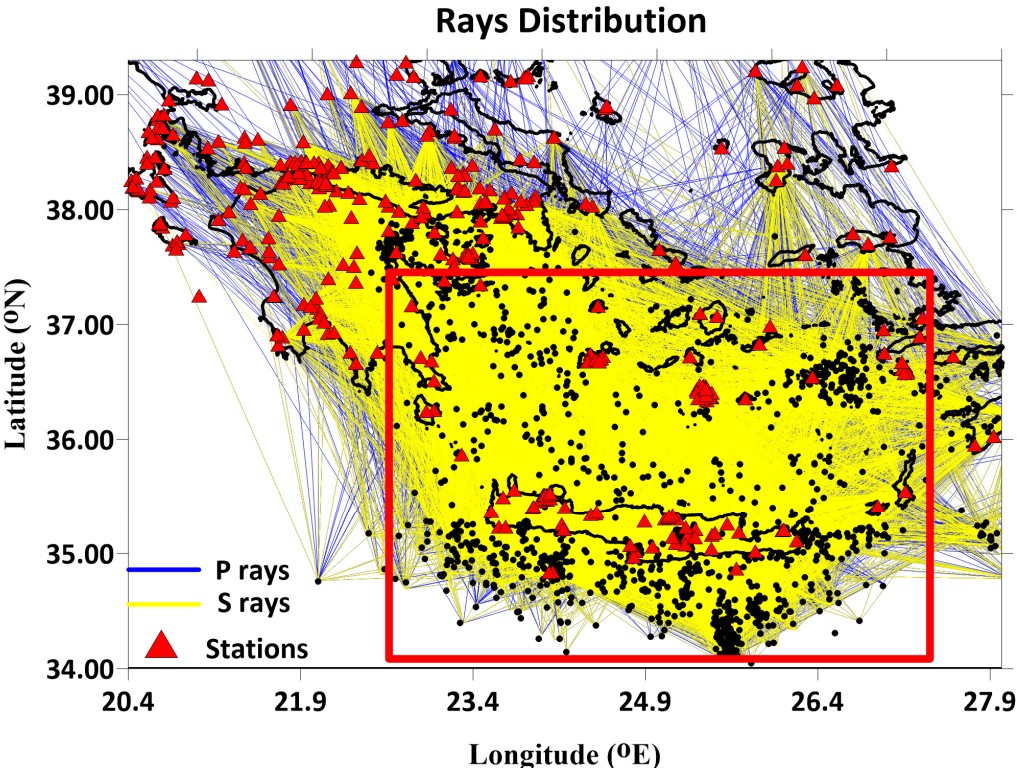

**Figure 4.** Total P- (blue) and S-ray (yellow) distribution. Red triangles indicate locations of the used stations. The selected seismicity used in this work (M ≥ 3.2) during 2018–2023 is presented by black dots. The red box represents the area of the southern Aegean, of which the tomography results will be presented in the following sections.

In this study, the inversion of body-wave traveltime data was based on the LOTOS scheme, as described by [38]. This code was previously used for studies of many different subduction zones with a similar geometry of observations and area dimensions [39–43]. For the inversion procedure, 1265 manually revised events located on the island of Crete and its neighboring regions were included during the first step. The dataset comprised 20587 P and 15370 S arrivaltimes, with at least 15 phases per earthquake (at least 10 for P- and 5 and for S-picks, respectively) and a ratio of S to P residual smaller than 1.5 (Figure 4). The inversion algorithm provides two different choices: a) inversion for both $V_P$ and $V_S$ ($V_P$–$V_S$ scheme) using the respective traveltime residuals ($dt_P$ and $dt_S$), and b) inversion for $V_P$ and $V_P/V_S$ ratio ($V_P$–$V_P/V_S$ scheme) using $dt_P$ and differential residuals, $dt_S$–$dt_P$. In the current study, tomographic inversion was applied for both schemes in order to obtain more information on the body-wave anomalies [38,44].

The calculations began with locating the sources in the reference 1D velocity model [35] using the grid search method. Based on the arrival times of P and S waves, the code performs iterative calculations and derives the 3D distributions of P and S velocities and the distribution of local events. We refined the grid successively to search for the best source location using three grids with 15 × 15 × 15 km, 5 × 5 × 5 km, and 1 × 1 × 1 km spacing We calculated an objective function for each grid point, representing the probability of the source location in the current point. Once we reached the extremum value of the objective function in a coarser grid, we performed a further search in a finer grid. Each iteration includes a source location, matrix calculation, and inversion. The sources werethen located with the updated 3D velocity models. During the execution of the algorithm, travel times

and ray paths werecalculated within the iteration by a 3D ray tracing sub-program of LOTOS that is based on the bending method as proposed by [45].

The velocity models were parameterized using a set of nodes deployed according to the ray density. In the horizontal view, the nodes were installed regularly with a spacing of 5 km. In the vertical direction, the grid spacing was inversely proportional to the ray density, but the spacing could not be smaller than a predefined value of 10 km. The results were calculated using four grids (with orientations of 0°, 22°, 45°, and 67°) and then averaged in one 3D model. The inversion was performed simultaneously for the P- and S-wave velocities, as well as for the source parameters (dx, dy, dz, and dt) and the station corrections. We used the LSQR algorithm for the inversion by [46,47]. The inversion stability was controlled by damping and flattening. This procedure minimized the differences in the solutions from neighboring nodes. The flattening of the velocity anomalies was controlled by adding equations with two nonzero elements with opposite signs, which corresponded to all pairs of neighboring nodes. The optimal values of flattening parameters were determined using synthetic modeling that enabled the best recovery of the models. Note also that the number of iterations has a similar effect as damping changes. Therefore, we always used five iterations and only varied the flattening parameters. To derive the final models, we used five iterations in total.

Sensitivity analysis for the available dataset was performed by applying the checkerboard synthetic test. This method uses alternating anomalies of positive and negative velocity perturbations on an initial 1D gradient model evenly distributed in a checkerboard pattern throughout the model. The stability of the inversion is controlled by amplitude damping and flattening regularization. The inversion of the large sparse matrix was performed using least squareswith QR decomposition (LSQR), while the optimal values of P- and S-wave amplitude damping, and smoothing as well as the station, source coordinates, and origin time corrections, were assessed based on the results of synthetic modeling as it is described in [38,48,49].

## 3. Results

The available tomographic software necessitates as input data the longitude and latitude of the available network stations and the arrival times from the recorded seismicity. The coordinates of the hypocenter and the time of origin are optional, as they are determined during the implementation of calculations. However, if preliminary hypocentral locations are available, they can be utilized to reduce the processing time of the operations. In addition, a generic 1D velocity model and a set of input parameters for performing the convergence iteration steps, comprising parameterization, grid dimensions, and damping parameters, were considered [38]. A 1D starting velocity model was set based on the one derived by [35]. This model was then optimized by running the complete tomographic procedure several times by calculating the average velocities at several depths and using them as a new starting model in the next phase of complete tomographic inversion (Figure 5). The model parameterization of the velocity field should be able to delineate, according to the local characteristics, the shape and perspective of heterogeneities. A nodal representation was used, since the velocity field reconstructed by a three-dimensional grid does not adopt any specific geometry of heterogeneities [50–52]. The interval of the grid nodes was kept significantly smaller than the expected size of the anomalies (<20 km) to weaken the distortion of the resulting models due to the grid configuration (Figure 6).

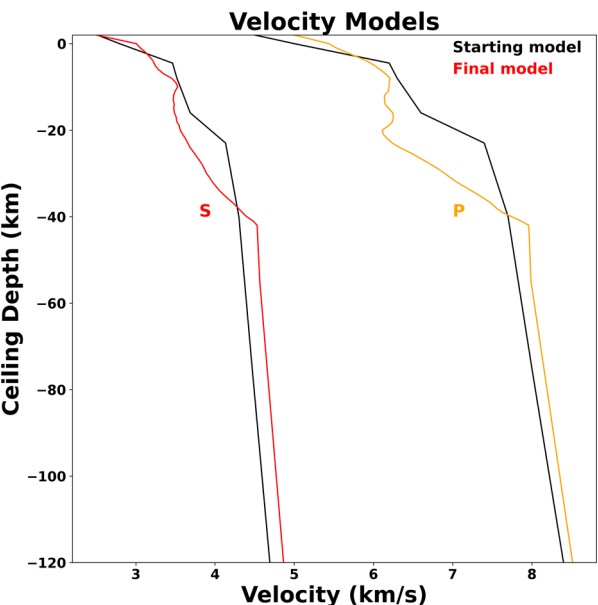

**Figure 5.** 1D reference model derived from the first stage of the procedure. Orange line indicates P-velocity, red one indicates S-velocity.

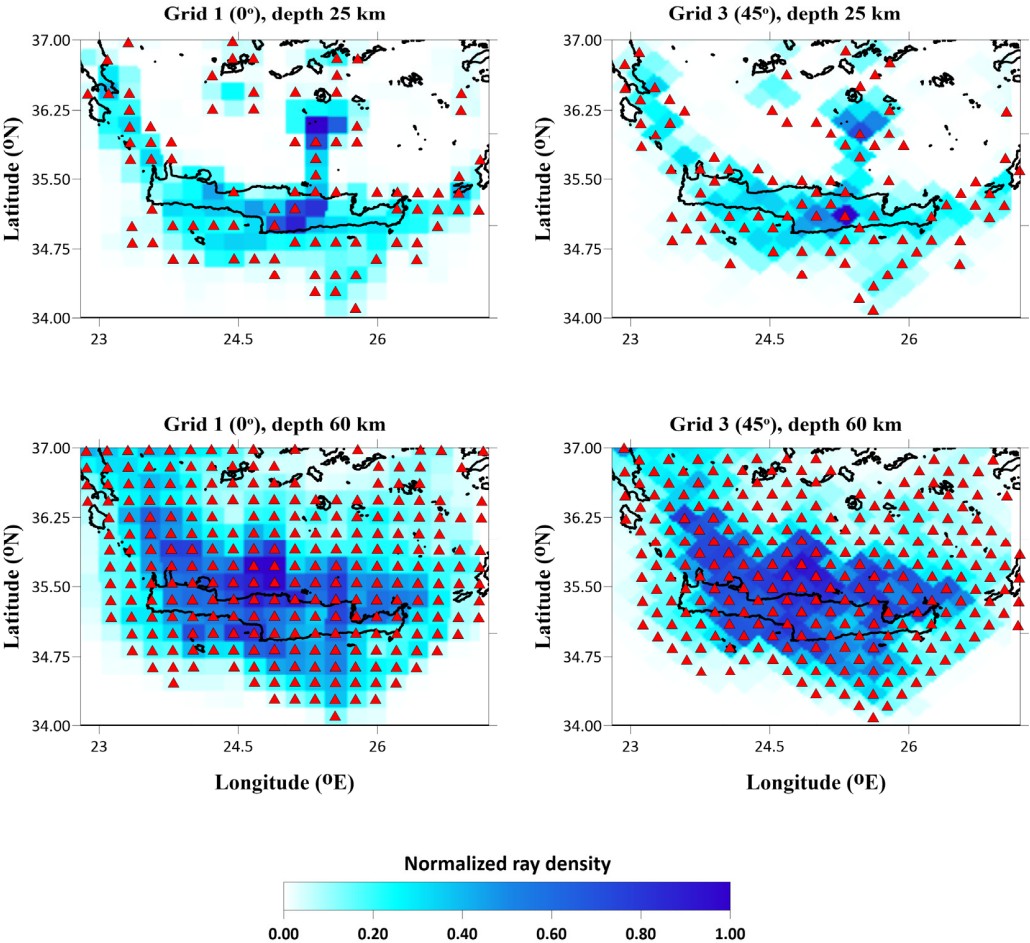

**Figure 6.** Normalized ray density of the P-waves with respect to the maximum value and nodes (red triangles) for two grids with the orientations of 0° and 45° at depths of 25 km and 60 km.

The optimal grid mesh was determined considering the stations/events geometry. The results were aggregated via a 3D model of the absolute P- and S-wave velocities by

simple averaging. Examples of node distributions and ray densities for two of the grid orientations in different depths are presented in Figure 6.

In the vertical direction, the grid spacing was inversely proportional to the ray density, but it could not be smaller than a predefined value (10 km in our case; Figure 7).

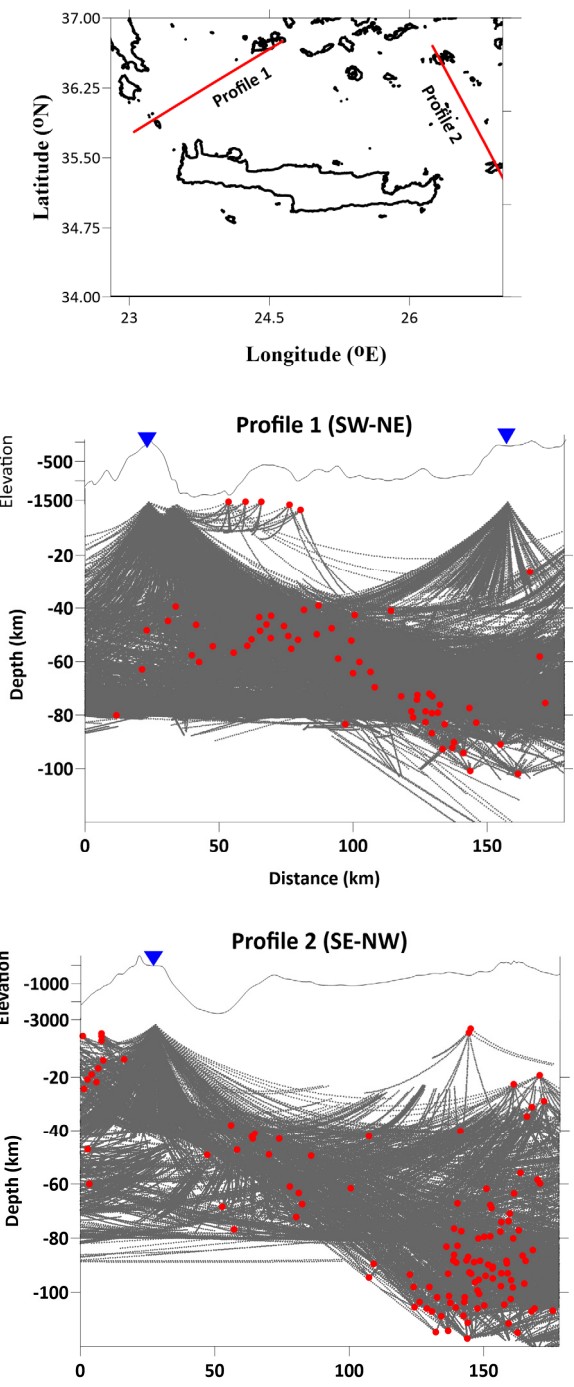

**Figure 7.** Vertical distribution of P-rays along the extension of profiles 1 and 2. Red dots indicate the position of the hypocenters. On the map of the upper row, there is the location of the two cross-sections. The width of each section is 4 km (±2 km from the centered line).

Figure 8 (left), the P- and S-wave traveltime residuals histogram before and after earthquake relocation, is shown. The initial P-wave travel time residuals were mainly distributed in the range of −2 to 1.8 s, and the average travel time residuals were 0.46 s, while the P-wave

travel time residuals were mainly distributed in the range of 0.6 s after relocation, and the average residuals were reduced to 0.27 s. Figure 8 (right) shows that the residuals of the initial S-wave travel time were also significantly reduced from 0.57 s to 0.28 s.

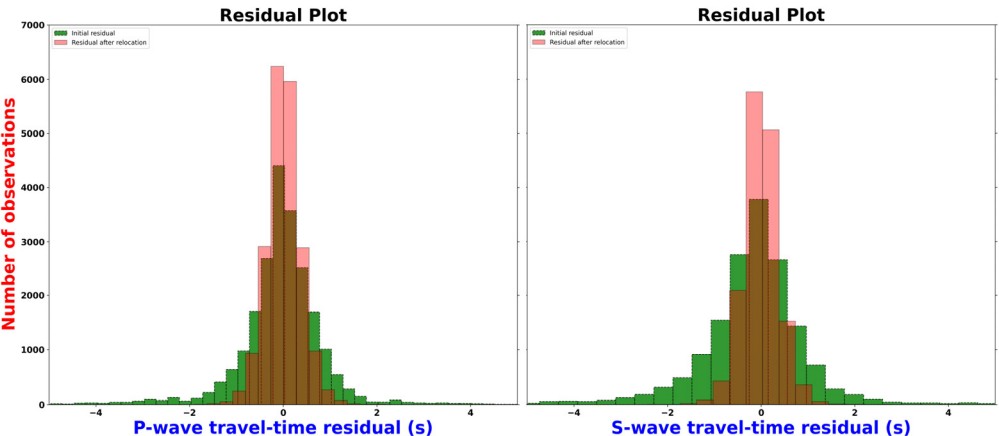

**Figure 8.** Histograms of P-wave (left) and S-wave (left) traveltime residualsbefore (green) and after (pink) the inversion procedure.

Moving on to Figure 8, we see a histogram of the P- and S-wave traveltime residuals before and after the earthquake relocation. The initial P-wave travel time residuals (left) were mainly distributed in the range of $-2$ to 1.8 s, and the average travel time residuals were 0.46 s. After relocation, the P-wave travel time residuals were mainly distributed in the range of $\pm 0.6$ s, and the average residuals were reduced to 0.27 s. On the right side of Figure 8, we see that the residuals of the initial S-wave travel time were also significantly reduced from 0.57 s to 0.28 s (Table 1). The overall seismic travel time residuals after relocation are mainly distributed in the range of $-2.2$ to 2 s. It appears that there has been a redistribution of the depths of the earthquake foci, with a majority of them now situated within a range of 0–25 km depth. However, it is worth noting that there are still significant concentrations of earthquake foci located below 50 km depth near the volcanic centers of the SAVA.

**Table 1.** Mean residuals in L1 norm and their variance reductions during the iterative inversion procedure.

| Iteration | P-Residual (s) | P-Residual Reduction (%) | S-Residual (s) | S-Residual Reduction (%) |
|---|---|---|---|---|
| 1 | 0.459 | 0.00 | 0.573 | 0.00 |
| 2 | 0.331 | 27.89 | 0.349 | 39.06 |
| 3 | 0.297 | 35.25 | 0.302 | 47.27 |
| 4 | 0.283 | 38.17 | 0.286 | 50.03 |
| 5 | 0.275 | 39.97 | 0.278 | 51.38 |

*3.1. Resolution Tests*

As a start, we present the results of the synthetic modeling. Besides verifying resolution limitations, such tests are essential for finding the optimal values of inversion parameters and evaluating noise's role in the data. In the LOTOS code workflow, the synthetic modeling follows the workflow of real data processing and simulates the same problems. Synthetic travel times were computed for the same source-receiver pairs as used in the final iteration of the real data inversion. At this step, we used the bending algorithm, which is briefly described in the methodology section. The execution of this sub-program conducted the tracing of the rays in the synthetic 3D model. The travel times were perturbed by random noise with a magnitude that has a similar variance reduction as theexperimental

data inversion. When starting the recovery of the model, we "forgot" about the true velocity model and perturbed the locations of sources. Then, we conducted the same inversion procedure as withthe experimental data analysis, starting from the sources' absolute locations in the starting 1D velocity model (Figure 5). Through synthetic tests, we were able to estimate the accuracy of the source locations during the inversion procedure. By comparing the final and the actual locations, we determined the level of uncertainty of the dataset. As shown in Figure 9, after five iterations the mean error in the source mislocations was reduced for both vertical profiles. It is worth noting that the source depth locations exhibited greater uncertainty.

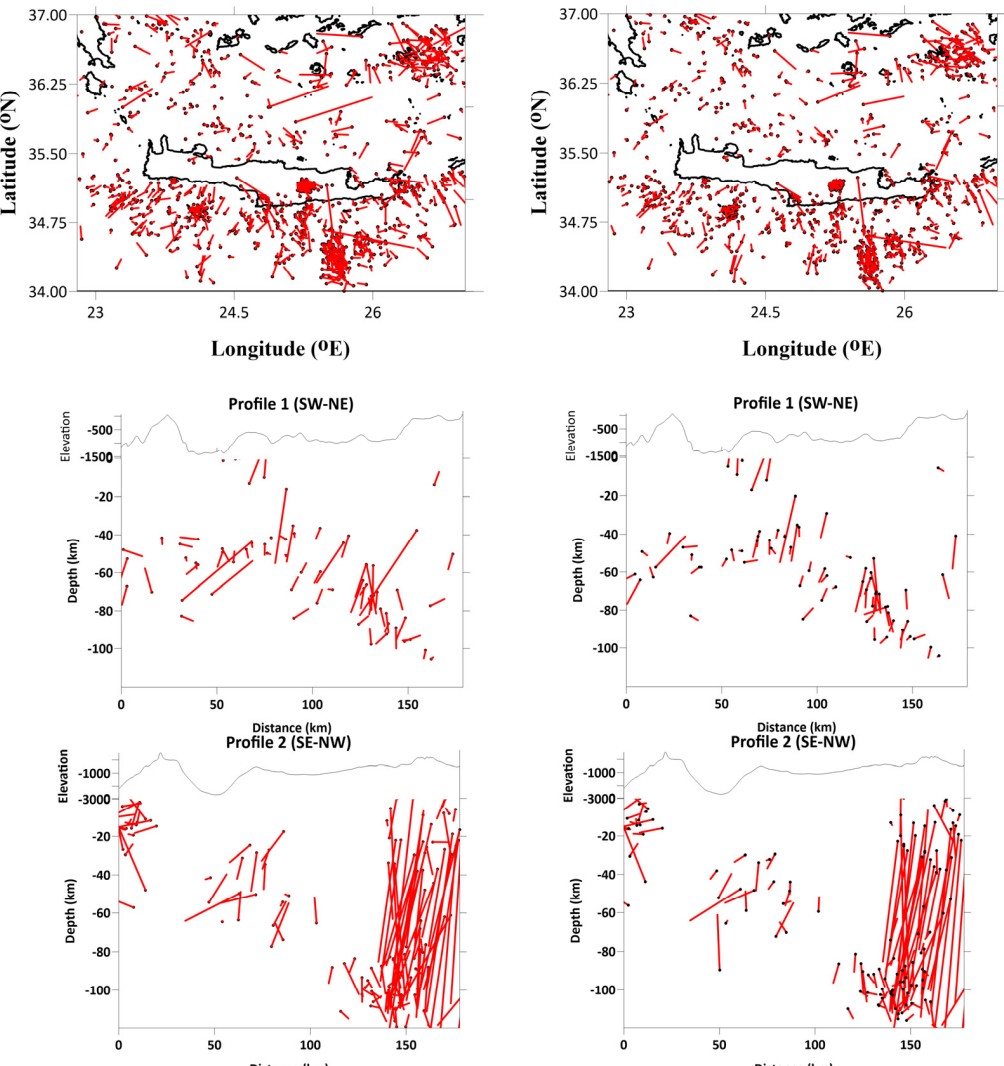

**Figure 9.** Mislocations of the sources during the synthetic modeling shown in horizontal slices and vertical sections. The first column shows the location results using the start of the 1D model, and the second column shows the location results of the final 3D velocity model. The black dots indicate the location of seismic events, and the red bars indicate the true locations. The figure was generated using the Golden Software Surfer.

To assess the horizontal resolution of the model, we performed a series of checkerboard tests [53] with anomalies of different sizes, as presented in Figures S1–S3. The adopted procedure included three different sets of dimensions of anomalies for the horizontal tests ($20 \times 20$ km$^2$, $40 \times 40$ km$^2$, and $80 \times 80$ km$^2$) in order to define the limitations of our model. The variations (%) of body-wave velocity anomalies ($\pm 10\%$) and the $V_P/V_S$ ratio distribution are presented in Figures S1–S3, at depths of 10, 40, and 80 km. Based on our

findings, it appears that the first set of anomalies with a $20 \times 20$ km$^2$ grid size is either not resolved or poorly resolved at various depth slices, whereas the larger sets with a $40 \times 40$ km$^2$ and $80 \times 80$ km$^2$ cell size provide reasonable resolution throughout most of the study area (in between Pliny Trench and the central-eastern part of SAVA). The $40 \times 40$ km$^2$ size pattern was particularly robustly resolved in most parts of the study area at shallower depths, up to around 40 km. However, at a depth of approximately 60 km, the $80 \times 80$ km$^2$ anomalies were more easily resolved than the smaller ones. Among the limits imposed by such a test is the fact that the extent of the near-vertical alteration of the structure is difficult to fully recognize due to a structure (checkerboard) where the diagonal elements in the vertical plane are strongly associated with dominant ray directions [53].

It seems that the models with smaller anomalies were not as clearly distinguished at deeper depth slices. It is worth noting that the anomalies of the 20 km board size were not well resolved for either the P- or S-wave velocity models, while the respective ones of 40 and 80 km were reasonably restored at all depth intervals (10–80 km) in the major part of the study area (34.75–35.9°N, 23.0–26.1°E). The results of the checkerboard tests suggest that there is a horizontal smearing towards the NE part of the study area due to a lack of seismological stations of the regional network of HUSN, and as a result there is a much smaller amount of reliable (accurately located) earthquake data.

To estimate the vertical resolution, we performed a synthetic test with anomalies defined along vertical profiles (Figure S4). Based on the patterns that we needed to investigate, we began setting the size of the checkerboard anomalies. We began with a board of 20 km in the horizontal direction, while in the vertical one the signs of the anomalies changed at depths of 20, 40, 60, and 80 km. For the shallow part of the model, the recovery result of this test appeared to be robust; the vertical resolution was sufficient to detect the change in the anomaly sign at depths of 20, 40, and 60 km, but the model did not have a sufficient resolution to recognize anomaly sign changes at 80 km. From the performed cross-section, it becomes evident that the anomalies at the ends of the profiles were smeared due to the poor data coverage along the SW and NE margins of the study area. This results from the poor azimuthal coverage between available stations and seismic events during the inversion procedure. It is interesting to note that the transition between negative and positive velocity anomalies is reconstructed sufficiently for the main part of the study area in profile 1 (west), with some horizontal smearing along the ray paths towards the NE. Profile 2 (east) appears to be less resolved than profile 1, particularly in the northern part of the section and for depths below 60 km. It is possible that the lower vertical resolution compared to the respective horizontal one may be due to a source and velocity parameters trade-off, as suggested by previous studies [54]. The synthetic tests we conducted showed that the absolute amplitudes of the body-wave anomalies were up to 4–5% smaller than the respective ones of the starting checkerboard grid. Overall, the checkerboard model was recovered to a satisfactory level, despite the presence of errors and outliers.

### 3.2. Tomographic Inversion Results

The generated P- and S-velocity anomalies (%) and the $V_P/V_S$ ratio distribution are depicted in four horizontal slices at 10, 25, 40, and 80 km depths (Figures 10 and 11) and two cross-sections (profile 1 and 2) perpendicular to the western and eastern part of the HT (Figure 12). The evaluation of the acquired values in both the horizontal slices and vertical profiles is restricted to model portions with a sufficient reconstruction of the checkerboard (Figures S1–S4). For P- and S-waves, velocity perturbations in the shallow depth layers range between −10 and +10%, while the $V_P/V_S$ ratio varies between 1.66 and 1.98 (Figure 11).

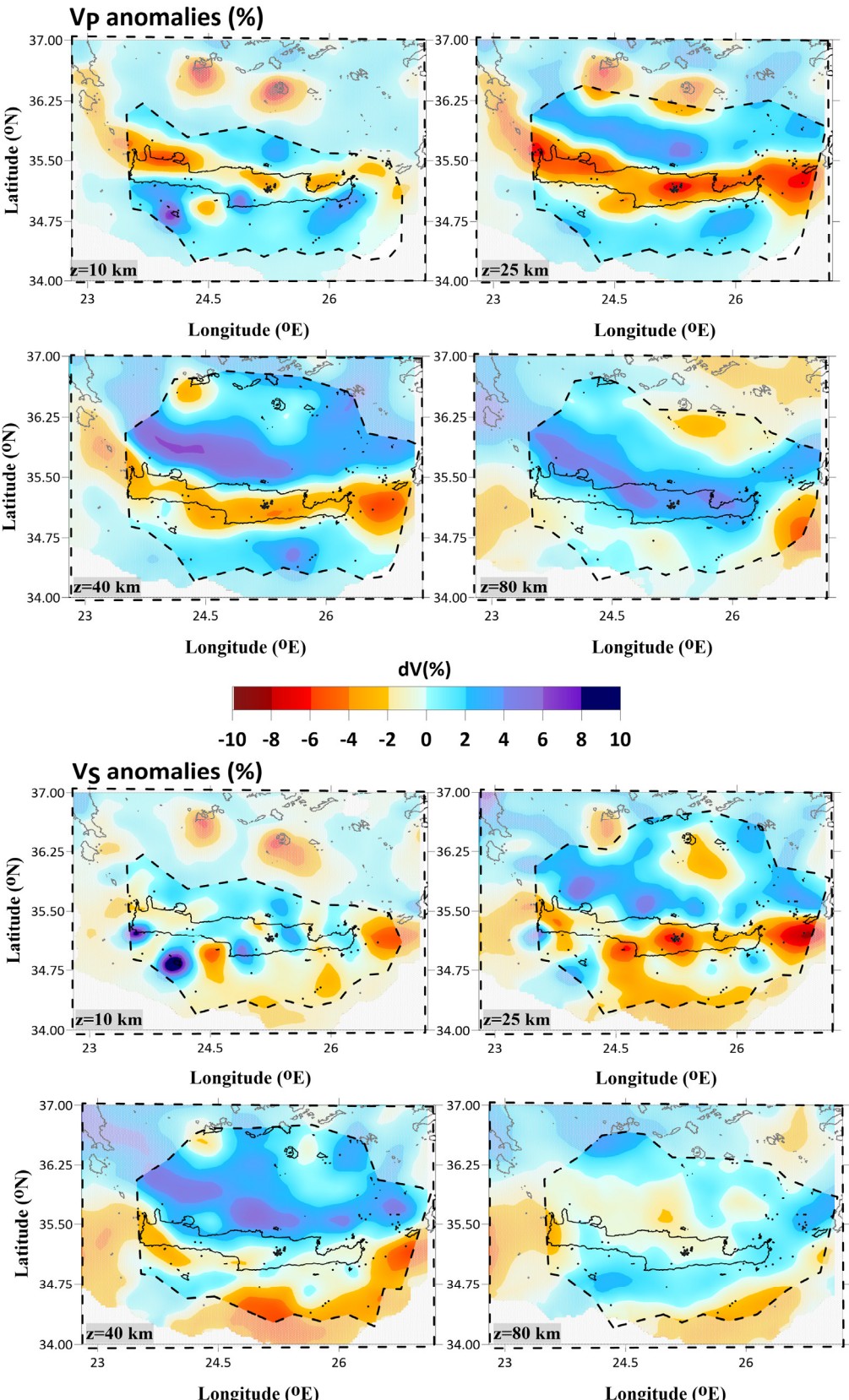

**Figure 10.** Tomograms of lateral $V_P$, $V_S$ (%) variations at 10, 25, 40, and 80 km depths. Areas with lower resolution are masked. Black circles indicate the recorded seismicity. Toponyms as in Figures 1 and 2.

According to synthetic modeling tests, the velocity structure is resolvable for depths ranging from 25 to 60 km and fairly reconstructed at the 80 km depth for the central part of the area of study (34.75–36.15°N, 23.2–26.25°E). This fact, together with intense ray coverage down to a depth of 40 km (Figure 6), lends confidence to the interpretation of the 3D inversion results.

It is interesting to note the prominent features identified by the present tomographic model, particularly the continuous positive velocity anomalies (%) located north of Crete. This pattern was also identified in previous studies [25,55–57], lending credibility to the model's accuracy. These positive velocity perturbations (%) are mainly attributed to the subduction of the Tethys oceanic lithosphere beneath the Aegean, a conclusion that is consistent with previous research (Figure 12). Additionally, it is worth noting that the front of the Hellenic Subduction Zone (HSZ) slab appears to be parallel to the HT in the region between Chania and Antikythera. Finally, it is intriguing that the positive velocity anomalies (%) are distributed in an amphitheater shape that is almost parallel to the HSZ, with the area of the positive anomalies increasing with depth.

Based on the results of this study, there are some noteworthy discontinuities, highlighted by the slow and fast body-wave (P, S) velocity anomalies towards the western and central-western regions of Crete, specifically within shallow depths (10–25 km), and there are some serious questions concerning the reliability of these results. It is plausible that this pattern is correlated with the local neotectonic faults striking in that direction, as illustrated in Figures 10 and 11. However, the limited resolution for anomalies smaller than $20 \times 20$ km$^2$ makes us skeptical about these results. Local tectonic studies have shown that these faults divide the Mesozoic Pindos Unit basement from the Neogene and Quaternary deposits [13,31], but the concentration of many seismic stations of both permanent and temporary networks near the plioseismal area of the Mw6.0 earthquake that occurred on 27 September 2021 and the large gap between the other stations that are on the island gives limited credibility to the interpretation of such results. A similar observation of body-wave velocity (P, S) anomalies is seen to the west, in the same depth slices (10 and 25 km) near the town of Chania, with negative and positive deviations from the reference 1D model to the west and east of Kissamos Gulf. This shallow, broad, negative $V_P$ anomaly (%) down to 25 km depth is common in collapsed zones due to the activation of normal faults.

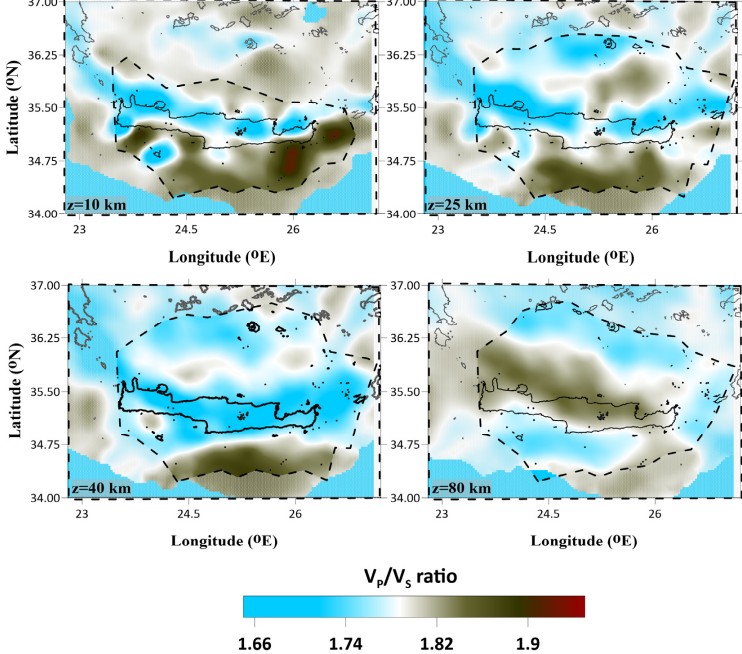

**Figure 11.** Tomograms of lateral $V_P/V_S$ ratio at 10, 25, 40, and 80 km depths. Areas with lower resolution are masked. Black circles indicate the recorded seismicity. Toponyms as in Figures 1 and 2.

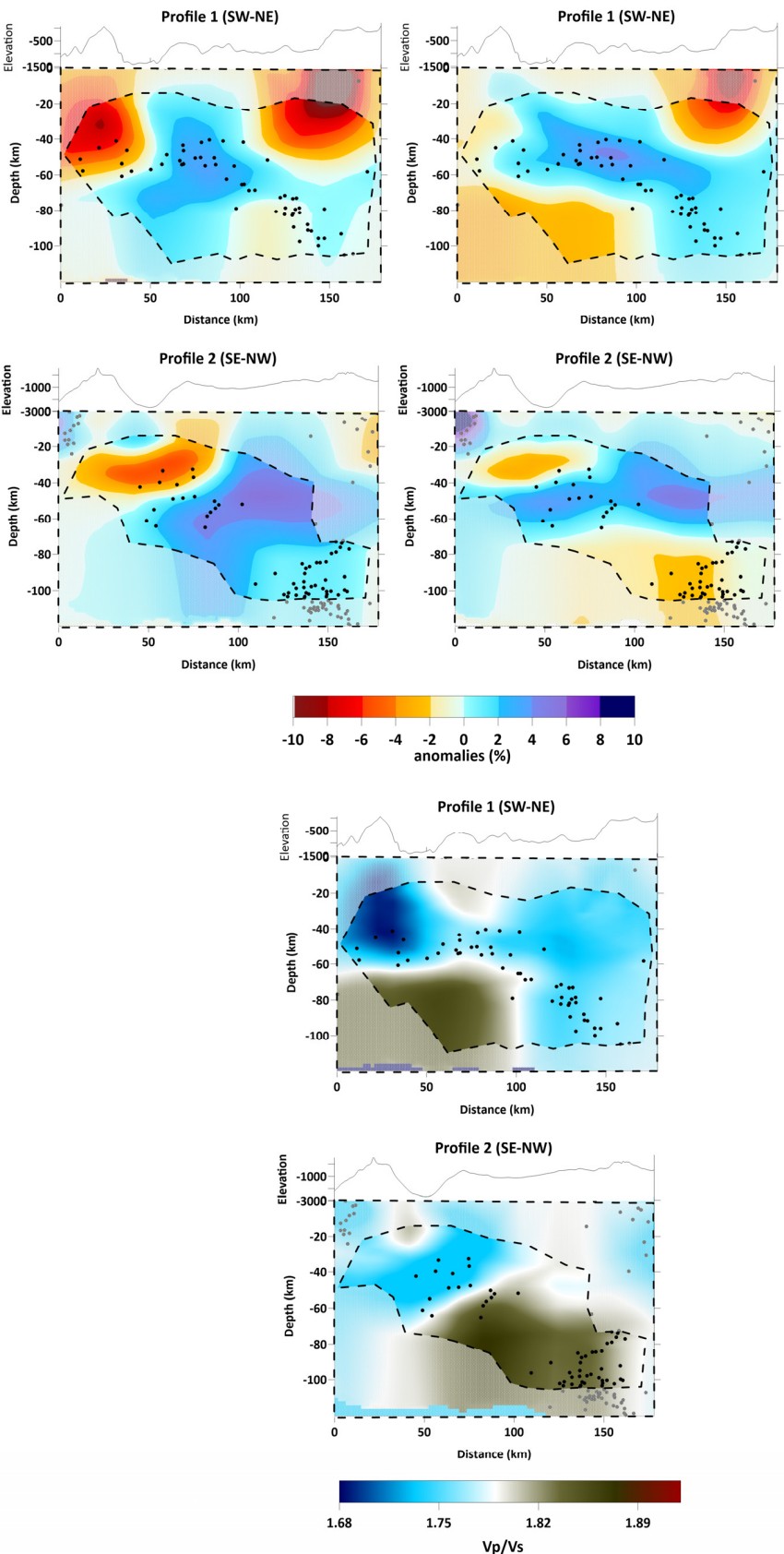

**Figure 12.** Distribution of $V_P$, $V_S$ (%) variations (left and central column) and $V_P/V_S$ ratio (right) in the performed cross-sections. Areas with lower resolution are masked. Black circles indicate the recorded seismicity. Map projection of cross-sections in Figure 7.

## 4. Discussion

The derived anomalies of body-waves ($dV_P$, $dV_S$) and the $V_P/V_S$ ratio provided important information about the southern Aegean regional tectonics and secondarily active faults of smaller scale (>20 km). In the study area, the seismic tomography model revealed a predominance of high-velocity anomalies north of Crete, in the region between Kythera and Karpathos, while close to the SAVA, the region is marked by significant low-velocity anomalies in the crust and uppermost mantle, beneath the active arc volcanoes. In the lower crust (25 km depth; Figure 10), along the SAVA, we observe a negative-velocity anomaly that could represent the magma feeding system beneath the Quaternary volcanic centers of Milos (36.7°N, 24.4°E) and Santorini (36.4°N, 25.4°E). The low ray density (Figure 6), the limited resolution for anomalies $20 \times 20$ km$^2$ and $40 \times 40$ km$^2$ (Supplementary Materials), and the non-uniform distribution between sources and stations indicate that there is limited credibility regardingthose observations and characterize them as artefacts.

The seismicity related to the HSZ is connected to high-velocity anomalies in the Sea of Crete and the islands south of SAVA. Its extension reaches up to 120–150 km (Figure 12), while the observed termination of the seismic activity in even greater depths may be due to plate dehydration embrittlement or runaway thermal shear stress, as highlighted by studies in similar environments [58,59]. More specifically, the main anomalies that have been identified in the region are:

(a) A large zone of negative- and or neutral- (%) velocity anomalies ($dV_P$, $dV_S$) and a low $V_P/V_S$ ratio in the shallow part of the Hellenic island-arc area (Kythera–Crete–Karpathos, Figures 10 and 11). Low crustal average $V_P/V_S$ ratios are indicative of a dehydrated crust. This may be attributed to deep crustal enrichment in silica as [60] proposed for the case of the Okhotsk Sea.

(b) In the area north of Crete Island, the results show a zone of positive-velocity perturbations ($dV_P$, $dV_S$) and a low $V_P/V_S$ ratio in the tomograms of 10, 25, and 40 km depths that can be correlated to the subducted Mediterranean oceanic lithosphere. The co-existence of high body-wave velocities (P, S) and a low $V_P/V_S$ ratio may be interpreted as the closure of subducted crust micro-cracks due to high pressure, as [60,61] have suggested. This observation is highlighted by profile 1 (Figure 12), where we mark the upper limit of the western segment of the HSZ slab as a positive-velocity perturbation (%) feature dipping towards the NE. This cross-section shows that the slab's dip decreases from ~60° (<40 km) to ~30-40° in depths greater than 70 km. In Figure 13, the positive anomalies related to the slab are consistent with the slab contours derived from Slab2.0 subduction zones [62].

(c) A region of slow (negative) body-wave velocity anomalies and a high $V_P/V_S$ ratio (~1.85) in the SAVA may reflect a hot upwelling flow related to slab dehydration, causing arc magmatism and volcanism in this region [44,47,62,63]. We have to keep in mind that the limited resolution for anomalies smaller than $20 \times 20$ km$^2$ and $40 \times 40$ km$^2$ and the low ray density in this area is a discouraging factor regarding the reliability of these results, as mentioned in the first paragraph of the section.

(d) In shallow depths (10 km) near the island of Crete, some anomalies of smaller size have been noticed. The presenceof negative body-wave (P, S) velocity anomalies and a low $V_P/V_S$ ratio (Figures 10 and 11; Heraklio, Central Crete: 35.15°–35.40°N, 25.05°–25.35°E) are observed. The distribution oflow $V_P/V_S$ ratio (~1.70-1.73)values maybe linkedto the fluid saturation of the surface neotectonic faults in this area [31,64], as recent studies have shown in regions of similar tectonics [65]. However, we must be cautious about this interpretation despite the resolution for anomalies smaller than $20 \times 20$ km$^2$ and the high P, S ray density in the area.

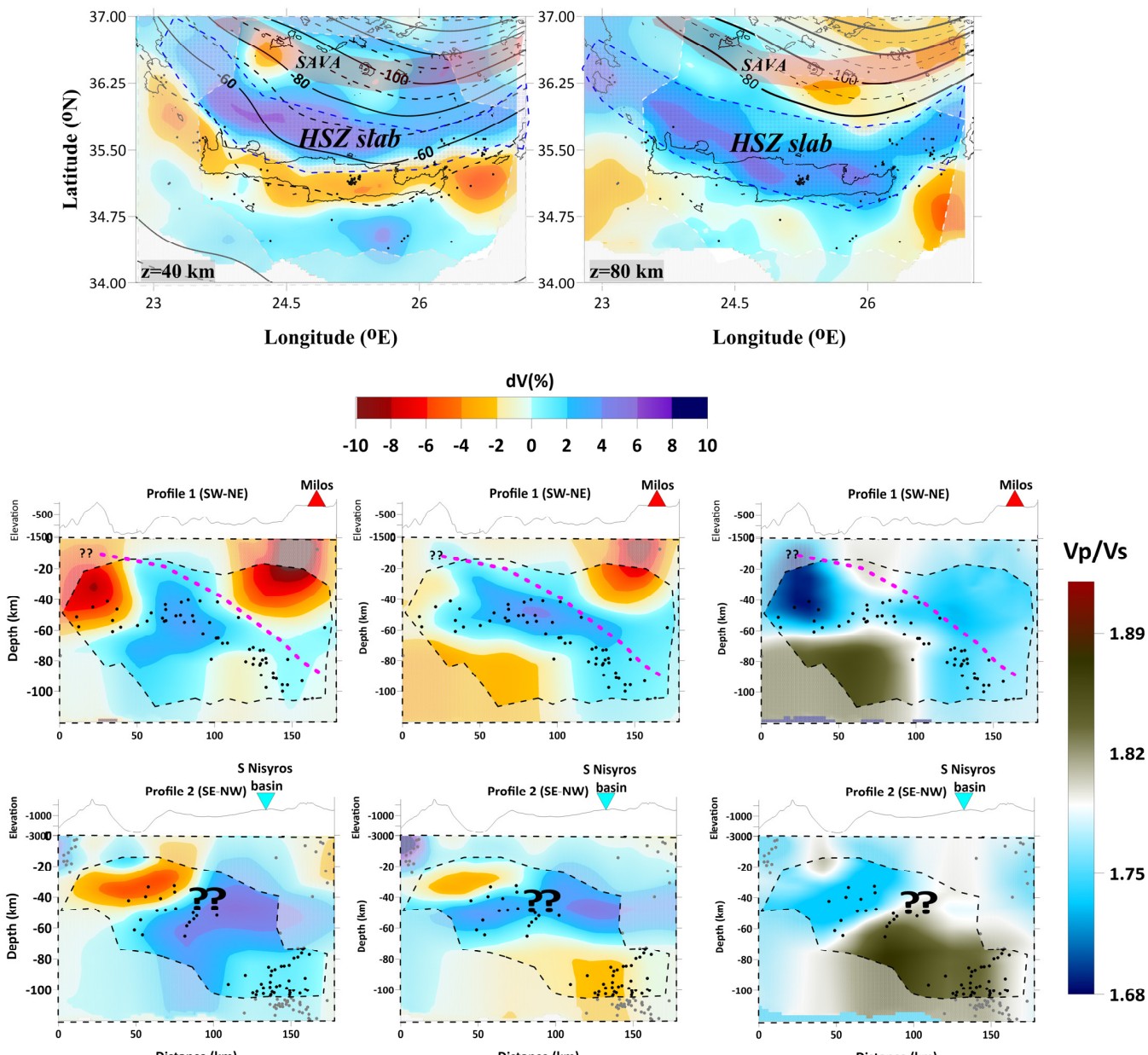

**Figure 13.** Interpretation of the results ofthe real-data inversion of $dV_P$ and $dV_S$ (%) tomograms (40 and 80 km depths), $V_P$, $V_S$ (%) variations (left and central column), and $V_P/V_S$ ratio (right) in the performed cross-sections, examining the regional-scale anomalies. Areas with lower resolution are masked. Black circles indicate the recorded seismicity.Black dashed lines on the tomograms of 40 and 80 km depths (first row) show the slab contours, derived fromSlab2.0 subduction zones [62]. Abbreviations-HSZ: Hellenic Subduction Zone; SAVA: South Aegean Volcanic Arc.

## 5. Conclusions

Based on the travel times of the manually revised events from 2018 to 2023, we obtained a new high-resolution 3D $V_P$ model of the crust and upper mantle on a regional scale for the region of the southern Aegean. The resulting 3D velocity models aimed to facilitate the improvement of the hypocentral parameters of the seismic events that occurred in this area. This process served as a springboard to a more detailed image of velocity anomalies that were obtained in the following stage of the tomographic inversion.

Negative body-wave (P, S) velocity perturbations, as well as the low $V_P/V_S$ ratio, support the existence of a dehydrated crust in superficial depthsin the Hellenic island-arc area (Kythera–Crete–Karpathos, Figures 10 and 11).

The regional distribution of seismic activity, body-wave velocity anomalies (P, S), and $V_P/V_S$ ratio values revealed:

- A complex shallow (<10 km) structure in Crete's central region mainly attributed to the dense pattern of neotectonic faults due to slow body-wave (P, S) velocity anomalies (negative perturbations) and low $V_P/V_S$ ratio;
- A region of significant low-velocity anomalies in the crust and uppermost mantle, close to the SAVA, marked by the active arc volcanoes;
- The existence of a low-angle feature of positive $V_P$ perturbations (%) correlated withthe observed intermediate-depth seismicity (H $\geq$ 40 km) in this part of the study area. This result could be related to the diving HSZ slab.

Additionally, expanding the research areas will be useful in investigating the geometry of the Wadati–Benioff zone in the southern Aegean, the properties of the seismogenic layer, and comparisons ofthis data with past studies and other regions globally. It may also be worth exploring the possibility of a tear in the slab near Karpathos and Rhodes Island in the eastern portion of the HSZ.

**Supplementary Materials:** The following supporting information can be downloaded at: https://www.mdpi.com/article/10.3390/geosciences13090271/s1, Figure S1: Reconstruction of P-wave anomalies for the depth slices of 10, 40 and 80 km and cell size of 20 (upper panel), 40 (second panel) and 80 km (third panel); Figure S2: Reconstruction of S-wave anomalies for the depth slices of 10, 40 and 80 km and cell size of 20 (upper panel), 40 (second panel) and 80 km (third panel); Figure S3: Reconstruction of $V_P/V_S$ ratio for the depth slices of 10, 40 and 80 km and cell size of 20 (upper panel), 40 (second panel) and 80 km (third panel); Figure S4: Checkerboard tests for checking the vertical resolution. The locations of the initial synthetic anomalies are indicated with rectangles of $20 \times 20$ km$^2$.

**Author Contributions:** Conceptualization, A.K.; methodology, A.K. and F.V.; software, A.K.; validation, A.K. and F.V.; formal analysis, A.K. and F.V.; investigation, A.K. and F.V.; resources, A.K. and F.V.; data curation, A.K. and F.V.; writing—original draft preparation, A.K.; writing—review and editing, A.K. and F.V.; visualization, A.K.; supervision, A.K. All authors have read and agreed to the published version of the manuscript.

**Funding:** This research received no external funding.

**Institutional Review Board Statement:** The study did not require ethical approval.

**Informed Consent Statement:** Not applicable.

**Data Availability Statement:** All data products generated in this study (velocity models, earthquake catalogues) are available from the authors upon request.

**Acknowledgments:** We would like to thank the scientists, post-graduate students, and personnel who participated in the installation or maintenance of the stations belonging to the HUSN and assisted inthe signal processing and manual location of the recorded seismicity and the reviewers for their creative comments and suggestions.We used data from the following seismic networks:the HL (Institute of Geodynamics, National Observatory of Athens, doi: 10.7914/SN/HL), HP (University of Patras, doi: 10.7914/SN/HP), HT (Aristotle University of Thessaloniki, doi: 10.7914/SN/HT), HA (National and Kapodistrian University of Athens, doi: 10.7914/SN/HA), HC (Seismological Network of Crete, doi:10.7914/SN/HC), and HI Institute of Engineering Seismology and Earthquake Engineering, doi: 10.7914/SN/HI) networks.

**Conflicts of Interest:** The authors declare no conflict of interest.

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
