# Peer review of "3D Body-Wave Velocity Structure of the Southern Aegean, Greece"

_geosciences, doi:10.3390/geosciences13090271_

Round 1

Reviewer 1 Report

The paper shows the results of a 3D travel-time tomography performed with regional seismic arrays in the Aegean Sea and Northern Crete Island. The article is well-written and well presented and the results are interesting. I would recommend this paper for possible publication in Geosciences after major revisions are performed. 

My comments are the following, and I also attach the PDF paper with more comments:

1) I detect a bad understanding of fundamental concepts like ray tracing, tomographic inversion, the meaning of the Vp/Vs ratio, and how it is related to rock nature. In the results, they mention this parameter constantly but it is not clear in the text how, for example, a low Vp/Vs ratio is related to faulting in the zone.  What does a high ratio mean? Authors must explain also how the Vp and Vs anomalies obtained in their results are affected individually by the nature of the rock where the seismic waves are propagating. These individual values per se will affect the Vp/Vs ratio obviously, but authors must describe why a high Vp/Vs ratio is high or low and how the physics of the rock affects this ratio (heat, fluids, brittle, etc).

2) The method section is a disaster. I think that the authors only read the manual of the program LOTUS and refer to a BSSA paper (Koulakov, I., 2009). I think authors could explain briefly how a travel-time tomography is prepared, how the ray-tracing is computed (e.g. Um and Thurber, 1986 (?)), how the underlying grid is established (author's criterion), and how the inversion for velocity is performed and not only saying the LOTUS program uses LSQ or so to get the tomography. Explain briefly the inversion method, if it was used LSQ say what damping you used to stabilize the inversion. You know that this kind of matrix inversion is unstable, so an inversion method must be used. Please explain explicitly which one LOTUS uses. I have the feeling that authors are using this software as a black box. Change the methodology section, please. 

3) Figures look more like a computer icon than a figure, they are so small that I can barely distinguish the different color anomalies described. Make the tomographic profiles or slices bigger. Labels and numbers embedded in the figures are so small that magnifying glass is needed.

4) I wrote more annotations in the article text in the PDF.

5) I want to check the paper again after corrections.

Greetings. 

Author Response

Questions of Reviewer #1 are marked in red color, while the answers in black.

1) I detect a bad understanding of fundamental concepts like ray tracing, tomographic inversion, the meaning of the Vp/Vs ratio, and how it is related to rock nature. In the results, they mention this parameter constantly but it is not clear in the text how, for example, a low Vp/Vs ratio is related to faulting in the zone.  What does a high ratio mean? Authors must explain also how the Vp and Vs anomalies obtained in their results are affected individually by the nature of the rock where the seismic waves are propagating. These individual values per se will affect the Vp/Vs ratio obviously, but authors must describe why a high Vp/Vs ratio is high or low and how the physics of the rock affects this ratio (heat, fluids, brittle, etc).

Both text of the main body and Figures of the manuscript were modified according to the pdf attached.

2) The method section is a disaster. I think that the authors only read the manual of the program LOTUS and refer to a BSSA paper (Koulakov, I., 2009). I think authors could explain briefly how a travel-time tomography is prepared, how the ray-tracing is computed (e.g. Um and Thurber, 1986 (?)), how the underlying grid is established (author's criterion), and how the inversion for velocity is performed and not only saying the LOTUS program uses LSQ or so to get the tomography. Explain briefly the inversion method, if it was used LSQ say what damping you used to stabilize the inversion. You know that this kind of matrix inversion is unstable, so an inversion method must be used. Please explain explicitly which one LOTUS uses. I have the feeling that authors are using this software as a black box. Change the methodology section, please.

The software is LOTOS. The version that was used is the revised one, LOTOS-12. More text was added both in the Methodology section as well as in the description of the results (synthetic and real-data inversion) according to the comments of the pdf attached. The backbone of the study was designed according to the recent research formats that the authors using this software have included in their studies. Keep in mind that we had to modify extensively the text due to the main gaps that specific plagiarism-checker software journals have that force us to change specific technical terms (trying not to copy from our past publications) related to the program we use (LOTOS in our case).  

3) Figures look more like a computer icon than a figure, they are so small that I can barely distinguish the different color anomalies described. Make the tomographic profiles or slices bigger. Labels and numbers embedded in the figures are so small that magnifying glass is needed.

Figures with horizontal slices were split into two as Reviewer #2 also suggested and the respective of cross-section were enlarged. The resolution is 300 dpi as according to the journal policy and the size is maintained in a reasonable dimension in order to fit the main body of the manuscript.

4) I wrote more annotations in the article text in the PDF.

The correction in the manuscript follows the pdf guidelines that was attached in the first round.

5) I want to check the paper again after corrections.

The revised manuscript is uploaded

Reviewer 2 Report

In this paper the authors describe the results of a velocity tomography analysis performed for an area of the Hellenic Arc. The paper is generally well written and well organized, the topic is appropriate for the Journal and results are interesting for seismologists interested to the study area. I have some concerns about discussion, conclusions and some figures, as described below. I think the paper may be appropriate for publication after some revision.

I would give more emphasis to the results than the sensitivity test. Figure 15 should be splitted in two figures to show better the result of the analysis, while figures 11, 12, 13, 14 could go in appendix.

I do not understand what is the difference between relocated hypocenters and starting locations. It is non visible in Figure 8. From the text it is not clear if and how important such difference is. Also, I do not understand the result shown in Figure 10, what it is exactly, how it was obtained and its usage. I think the authors should revise and clarify this part of the paper.

The result of tomography analysis is not fully convincing. The low velocity anomalies shown in Figure 15 are all centered in the places where more seismic stations are available (compare Figure 15 with the red box in Figure 4). On the contrary, high velocity anomaly characterize the sea, north and south of Crete island, where no seismic stations were installed. This can not be a coincidence, and this feature should be discussed in the paper.

The result shown in Figures 16 and 17 for profile 2 vertical sections does not show any likely correlation with the expected lithospheric structure of the area. In fact question marks in Figure 17 profile 2 are appropriate!. In my opinion these plots indicate that the result of this tomography study is less reliable than what inferred from the sensitivity tests. This is not authors’ fault, it is simply what happens very often with this kind of studies due to the non-uniform distributions of sources and hypocenters. Probably a lot of tomography results are less reliable than peaple believe.

The discussion contains some speculative sentences, like for example at lines 390-392. Why the velocity anomalies should be related with the seismic sequence of Arkalochori 2021-2022, and how? Please, explain or erase that text. Also the sentence at lines 379-380 about the “… upward movement of fluids...” appears very speculative.

The sentence at line 430 about the seismic hazard is unappropriate for the conclusions and should be erased. The contribution of tomography to the evaluation of seismic hazard is negligible.

Figure 4. Please, cite the red box in the caption.

Figure 7. In this figure the ray density probably would give a more precise idea than the grey area entirely covered by rays. Some rays are very deep in the two sections. It is not clear to which sources-receivers they correspond. Are they correct? Please, check. What is the section thickness?

Figure 8. I can’t appreciate the content of this figure. Symbols in the two maps are slightly different (smaller, or with a thinner border in plot a), but the two distributions appear extremely similar and it is impossible to catch the difference. Red triangles representing seismic stations are hardly distinguished from red epicenters.

Figure 10 is difficult to read. Symbols are too small and lines too narrow.

The paper is well written. Language is mostly clear, although it can probably be improved by a mother tongue.

Author Response

Questions of Reviewer #2 are marked in red, while the answers of the authors in black.

I would give more emphasis to the results than the sensitivity test. Figure 15 should be splitted in two figures to show better the result of the analysis, while figures 11, 12, 13, 14 could go in appendix.

Figure 15 was split into two parts, one with body-wave perturbations and another one of Vp/Vs ratio. Figures 11-14 were moved in the appendix.

I do not understand what is the difference between relocated hypocenters and starting locations. It is non visible in Figure 8. From the text it is not clear if and how important such difference is.

Figure 8 was removed.

Also, I do not understand the result shown in Figure 10, what it is exactly, how it was obtained and its usage. I think the authors should revise and clarify this part of the paper.

A paragraph was added concerning this figure in order to clarify its usage (second paragraph in the section of resolution tests 3.1.)

The result of tomography analysis is not fully convincing. The low velocity anomalies shown in Figure 15 are all centered in the places where more seismic stations are available (compare Figure 15 with the red box in Figure 4). On the contrary, high velocity anomaly characterize the sea, north and south of Crete island, where no seismic stations were installed. This can not be a coincidence, and this feature should be discussed in the paper.

The section was modified, also according to the suggestions of Reviewer #1.

The result shown in Figures 16 and 17 for profile 2 vertical sections does not show any likely correlation with the expected lithospheric structure of the area. In fact question marks in Figure 17 profile 2 are appropriate!

Figure 17 is the interpretation of Figure 16. Both figures were upgraded for some features to be clearer after the report of Reviewer #1 also. We agree on the objection in profile 2 that was the reason we had to put question marks. A part was added in the text in order to make it more evident and justify this part.

In my opinion these plots indicate that the result of this tomography study is less reliable than what inferred from the sensitivity tests. This is not authors’ fault, it is simply what happens very often with this kind of studies due to the non-uniform distributions of sources and hypocenters. Probably a lot of tomography results are less reliable than people believe.

A related text was added a) at the end of ‘resolution tests’ section b) page 15 in ‘tomographic inversion’ section and it is discussed in the following paragraph (Discussion).

The discussion contains some speculative sentences, like for example at lines 390-392. Why the velocity anomalies should be related with the seismic sequence of Arkalochori 2021-2022, and how? Please, explain or erase that text.

The sentence was erased.

Also the sentence at lines 379-380 about the “… upward movement of fluids...” appears very speculative.

A text was added on this particular matter.

The sentence at line 430 about the seismic hazard is unappropriate for the conclusions and should be erased. The contribution of tomography to the evaluation of seismic hazard is negligible.

The sentence was removed

Figure 4. Please, cite the red box in the caption.

The red box was cited in the caption

Figure 7. In this figure the ray density probably would give a more precise idea than the grey area entirely covered by rays. Some rays are very deep in the two sections. It is not clear to which sources-receivers they correspond. Are they correct? Please, check. What is the section thickness?

The figure and its caption was re-designed, re-written and put into the manuscript. The thickness of both sections is equal to 4km (±2 km from the red line/section).

Figure 8. I can’t appreciate the content of this figure. Symbols in the two maps are slightly different (smaller, or with a thinner border in plot a), but the two distributions appear extremely similar and it is impossible to catch the difference. Red triangles representing seismic stations are hardly distinguished from red epicenters.

Figure 8 was removed from the final version

Figure 10 is difficult to read. Symbols are too small and lines too narrow.

Figure was re-sized in the updated version.

Reviewer 3 Report

This is a well-written paper, which proposes a more detailed image of velocity anomalies based on a new high-resolution 3-D Vp model of the crust and upper mantle on regional scale for the region of the Southern Aegean. The tomographic inversion results and discussion are impressive. My suggestions are as follows:

(1) For the Data and Method, it is better to add a graph that is more clear for readers.

(2) In the Data and Method, it is better to extend the dataset and add the number of available stations, enhancing confidence in results.

(3) In the results, it is better to add Profile 3 (E-W) below Crete Island, analyzing the partial characteristics of velocity anomalies.

(4) In later research, expanding the research areas will be useful in investigating the geometry of the Wadati-Benioff zone in Southern Aegean, the properties of the seismogenic layer.

Author Response

The questions of Reviewer #3 is marked in red, while the answers in black color.

(1) For the Data and Method, it is better to add a graph that is more clear for readers.

The figures were re-designed according to the reviewers comments.

(2) In the Data and Method, it is better to extend the dataset and add the number of available stations, enhancing confidence in results.

The methodology section was filled with additional text in order to be more clear for the reader.

(3) In the results, it is better to add Profile 3 (E-W) below Crete Island, analyzing the partial characteristics of velocity anomalies.

This is really hard (if not impossible) due to a) the time limitation by the journal to upload a revised manuscript b) the absence of resolution and d) the lack of P, S rays in certain parts where there is a gap in the distribution of the seismic stations.

(4) In later research, expanding the research areas will be useful in investigating the geometry of the Wadati-Benioff zone in Southern Aegean, the properties of the seismogenic layer.

We will certainly, and we will need to fill the data with available stations from nearby national networks.

Round 2

Reviewer 1 Report

The new version improved with the revision. No more review is needed, and I recommend the paper for publication. I don't need to see the paper again.

Author Response

Dear Sir,

We would like to express our gratitude once more for the review and the marks that ameliorated the overall content of the manuscript.

Kind regards 

Reviewer 2 Report

Figure 10 shows the results of tomography analysis, thus it is the most important of the paper. I suggest to erase all lines that represent tectonic structures (they make non sense for sections deeper than a few km!) and leave only the coastline that help to figure out where anomalies are located.

The same suggestion for Figure 11.

I do not understand the very many (%) added in the text. I don’t think it is necessary.

English is quite good. It can be improved by a mother tongue in some parts of the paper.

Author Response

Figure 10 shows the results of tomography analysis, thus it is the most important of the paper. I suggest to erase all lines that represent tectonic structures (they make non sense for sections deeper than a few km!) and leave only the coastline that help to figure out where anomalies are located.

The same suggestion for Figure 11.

Figures were modified according to Reviewer #2 suggestions

I do not understand the very many (%) added in the text. I don’t think it is necessary.

We had to put them in the main text according to Reviewer #1 questions and suggestions in these specific parts.

Once more we would like to acknowledge your contribution to the upgrade of the manuscript.

Kind regards

Reviewer 3 Report

The revised version looks good to me. I have no further suggestions.

Author Response

Dear Sir,

We would like to express our thanks for the proposals to improve the overall content of the manuscript and concentrate on future work.

Kind regards